# When Pulmonologists Are Novice to Navigational Bronchoscopy, What Predicts Diagnostic Yield?

**DOI:** 10.3390/diagnostics12123127

**Published:** 2022-12-12

**Authors:** Louise L. Toennesen, Helene H. Vindum, Ellen Risom, Alexis Pulga, Rafi M. Nessar, Arman Arshad, Alice Christophersen, Yoon Soo Park, Kristoffer Mazanti Cold, Lars Konge, Paul Frost Clementsen

**Affiliations:** 1Department of Pulmonary Medicine, Copenhagen University Hospital, 2650 Hvidovre, Denmark; 2Department of Onchology, Aarhus University Hospital, 8200 Aarhus, Denmark; 3Department of Pulmonary Medicine, Odense University Hospital, 5000 Odense, Denmark; 4Department of Pulmonary Medicine, Bispebjerg University Hospital, 2400 Copenhagen, Denmark; 5Department of Pulmonary Medicine, Zealand University Hospital, 4000 Roskilde, Denmark; 6Massachusetts General Hospital, Boston, MA 02114, USA; 7Harvard Medical School, Boston, MA 02115, USA; 8Faculty of Health and Medical Sciences, University of Copenhagen, 1165 Copenhagen, Denmark; 9Copenhagen Academy for Medical Education and Simulation (CAMES), Rigshospitalet, University of Copenhagen and the Capital Region of Denmark, 1165 Copenhagen, Denmark

**Keywords:** lung cancer, diagnostic equipment, diagnosis, invasive procedures, education

## Abstract

Predicting factors of diagnostic yield in electromagnetic navigation bronchoscopy (ENB) have been explored in a number of previous studies based on data from experienced operators. However, little is known about predicting factors when the procedure is carried out by operators in the beginning of their learning curve. We here aim to identify the role of operators’ experience as well as lesion– and procedure characteristics on diagnostic yield of ENB procedures in the hands of novice ENB operators. Four operators from three centers without prior ENB experience were enrolled. The outcome of consecutive ENB procedures was assessed and classified as either diagnostic or non-diagnostic and predicting factors of diagnostic yield were assessed. A total of 215 procedures were assessed. A total of 122 (57%) of the ENB procedures resulted in diagnostic biopsies. Diagnostic ENB procedures were associated with a minor yet significant difference in tumor size compared to non-diagnostic/inconclusive ENB procedures (28 mm vs. 24 mm; *p* = 0.03). Diagnostic ENB procedures were associated with visible lesions at either fluoroscopy (*p* = 0.003) or radial endobronchial ultrasound (rEBUS), (*p* = 0.001). In the logistic regression model, lesion visibility on fluoroscopy, but none of operator experience, the presence of a bronchus sign, lesion size, or location nor visibility on rEBUS significantly impacted the diagnostic yield. In novice ENB operators, lesion visibility on fluoroscopy was the only factor found to increase the chance of obtaining a diagnostic sample.

## 1. Introduction

It is crucial to achieve material for pathological evaluation of pulmonary lesions suspected of malignancy to establish correct treatment. There is considerable focus on assessing factors that influence the diagnostic yield of the invasive procedures used to diagnose these lesions. Electromagnetic navigation bronchoscopy (ENB) is a promising technology that is designed to guide the biopsy of peripheral pulmonary lesions, thereby increasing the diagnostic yield of conventional bronchoscopy [1]. ENB utilizes an electromagnetic technique for real-time navigation based on three-dimensional computed tomography (CT) scans and virtual bronchoscopy. Using computer software and a recent three-dimensional bronchial map from a chest scan, the operator is able to navigate to a desired location within the lung to biopsy lesions, stage lymph nodes, insert markers to guide radiotherapy or guide brachytherapy catheters [2]. 

It has been established that factors such as lesion size, radial endobronchial-ultrasound (rEBUS) and proximity to the hilum, as well as computed tomography (CT) bronchus sign (air-filled bronchus near the lesion as seen on CT), increase diagnostic yield in conventional bronchoscopy [3,4,5,6]. Although some studies have suggested that this is also the case in ENB [7,8] the conclusions have been based on data from operators with substantial prior experience in ENB. Very little is known about predicting factors and should be evaluated systematically [9,10], including peri-procedural findings, such as lesion visibility on radial EBUS or fluoroscopy, when operators with limited ENB experience perform the procedure. 

It is now broadly accepted that screening for lung cancer reduces cancer mortality [11]. For this reason, the pulmonologist can expect the referral of more patients with small lung lesions, and in line with this, training techniques and programs in endoscopic procedures for novice operators are gaining increasing interest [12,13]. However, especially concerning the more recent invasive endoscopic procedures on the market, such as ENB, there is still huge need for knowledge about what factors influence diagnostic yield when inexperienced operators perform ENB in order to select suitable patients and to decide when expert supervision is needed. 

We aimed to clarify predictors for diagnostic ENB procedures in ENB novices’ hands. 

## 2. Materials and Methods

### 2.1. Design

In an observational prospective design, four operators were enrolled from three hospitals in Denmark where the ENB equipment was purchased recently. We have previously published the study design as well as data of operators’ learning curves [14]. In short, we included all consecutive ENB procedures in consenting adult patients who were candidates for ENB. The indication of ENB was based on consensus between at least two doctors who were pulmonary endoscopists who reviewed CT, clinical status, lung function, etc., and presence or absence of other lesions that might be reachable with other procedures (for example EBUS or endoscopic ultrasound with the bronchoscope (EUS-B)), as well as the risk in connection with CT-guided biopsy as an alternative to ENB. Thus, the indication of ENB was not solely based on characteristics of the lung lesion. The ENB operator was not necessarily one of these doctors.

The inclusion period was from 1 May 2018 to 1 February 2020. If an operator had not performed at least 50 ENB procedures by the end of the inclusion period, we extended the inclusion period by an additional 6 months for the operator in question in order to include as many procedures as possible. However, due to time constrains and regulatory demands, it was not possible to extend the inclusion period any further, even if operators had not reached 50 ENB procedures. Due to local regulatory decisions, the ENB equipment was not acquired simultaneously at all three hospitals, and thus Operator 1 was enrolled for data-collection on 1 May 2018, Operator 2 was enrolled on 1 October 2018 and Operators 3 and 4 were enrolled on 1 August 2019.

### 2.2. Human Ethics Approval Declaration

All patients gave written informed consent to participate in the study, which was approved by the Danish Data Protection Agency (19-000079/2018-048). Since the study was an observational study without randomization or experimental procedures, it did not fall under the jurisdiction of the scientific ethics system (H-19040627).

### 2.3. Operators

The four operators were all experienced, board certificated pulmonary specialists who had performed between 1100 and 5000 conventional bronchoscopies and between 1000 and 2500 radial EBUS procedures.

Operator 1 had minor earlier experience with the ENB equipment, as he had performed 15 ENB procedures during previous employment in another hospital more than 5 years prior to his enrolment in the study. The remaining three operators had no prior experience with ENB. The pre-study training was identical for all operators and consisted of a two-day hands-on course provided by Medtronic (Minneapolis, MN, USA).

### 2.4. Procedures

All ENB procedures were performed using the SuperDimension^TM^ navigation system version 7.1 (Medtronic, Minneapolis, MN, USA). All additional procedural decisions, including choice of biopsy tools (forceps, needle, cytobrushes, bronchial lavage), order of biopsy tool use and use of fluoroscopy (confirmed with frontal images), were performed at the operator’s discretion. In all three centers, intravenous Midazolam and Fentanyl were used for sedation, and all procedures were performed using Olympus (190F) bronchoscopes, rEBUS probes were Olympus UMS20-20R and forceps were Olympus FB-231D.

The location of the lesions (central, middle, peripheral) were recorded based on the definition used in the NAVIGATE study [15], with peripheral lesions defined as lesions in the peripheral outer third of the lungs and the central lesions located in proximity to the hilum. 

All operators used rEBUS as a standardized part of the ENB procedure whenever possible. There was no access to rapid onsite evaluation of cytological material (ROSE). All procedures were performed under conscious sedation and any adverse events related to the procedures were recorded. None of the pulmonologists had access to supervision by experienced ENB operators. 

### 2.5. Diagnosis and Definition of Diagnostic Yield

The final diagnosis (malignant or non-malignant) was verified by reviewing patients’ electronical charts and their pathology reports six months after the ENB procedure was performed. Based on this, the outcome of each ENB procedure was classified as either a diagnostic procedure or non-diagnostic procedure based on sample adequacy (defined as described below). In case the operator was unable to guide the locatable guidewire to the lesion the procedure was considered a non-diagnostic procedure regardless of sample diagnosis. Diagnostic yield was defined as the number of diagnostic procedures relative to the total number of ENB procedures

Diagnostic procedures: Samples with malignant cells at cytopathological evaluation.Samples showing a specific benign diagnosis including either infection, granulomas or cryptogenic organized pneumonia at cytopathological and/or microbiological evaluation.Samples without a specific benign diagnosis and without malignant cells in examples samples with mild inflammation AND (for all cases) at least one subsequent CT-scan after at least 6 months with evidence of regression of the lesion.

Non-diagnostic procedures: Samples without a specific benign diagnosis and without malignant cells with no follow-up registration, for example, if patients had died or did not wish to attend clinical follow-up.Samples that are deemed inadequate at cytopathological evaluation.Samples without a specific benign diagnosis and without malignant cells where a CT-scan or a chest X-ray during the following six months show growth of the lesion.Samples without a specific benign diagnosis and without malignant cells where subsequent invasive procedures such as CT-guided biopsy or EBUS return with either a malignant or a specific benign diagnosis.All procedures where the final diagnosis was not based on a sample obtained by the use of ENB, but based on one or more samples obtained *in relation* to the ENB procedure using other equipment, such as rEBUS or conventional endobronchial biopsy under direct vision.

### 2.6. Definition of Complications

All procedure-related complications were recorded. Severe adverse events were defined as: (1) patient’s death during the procedure, (2) pneumothorax requiring drainage or (3) bronchopulmonary hemorrhage requiring a balloon catheter or blood transfusion.

### 2.7. Statistics

All data were analyzed using Stata 16 (College Station, TX, USA). Continuous variables were described using means and standard deviations (SD). Dichotomous variables were summarized in simple proportions. Statistical comparisons of continuous variables were performed using ANOVA, while those comparing proportions performed done using the Fisher’s Exact test. Continuous variables were checked for normal distribution before being used in the ANOVA. Univariate regression analyses were conducted to assess which factors were associated with diagnostic yield. Furthermore, cross-classified random effects logistic regression models controlling for clustering effects within operators and across lesion location were specified to assess the effects of each operator’s progressive ENB experience (repetition number) on diagnostic yield while adjusting for lesion and procedure characteristics, such as lesion size and location, presence of a bronchus sign and lesion visibility on fluoroscopy and rEBUS. Predictors selected in the regression model were based on model fit indices (likelihood ratio test and information criteria). A *p*-value of less than 0.05 indicated statistical significance.

## 3. Results

A total of 215 procedures were performed divided by operator as follows: 79 (Operator 1), 47 (Operator 2), 64 (Operator 3) and 25 (Operator 4) (Figure 1). A total of 122 (57%) of the ENB procedures resulted in diagnostic procedures (Figure 2).

A total of 147 (68%) of lesions were concluded to be malignant. Of these, 75 (51%) were diagnosed with ENB.

The number of lesions biopsied in each patient was one, except in four patients, in whom two lesions were biopsied. Pneumothorax requiring chest tube insertion occurred in two cases (0.9%) and bronchopulmonary hemorrhage requiring a balloon catheter occurred during one procedure (0.5%).

Most lesions were located in the right upper lobe (74 (34%)) and in the left upper lobe (61 (28%)) (Table 1). A total of 126 (59%) lesions were located in the peripheral third of the lungs, whereas 67 (31%) and 22 (10%) of the lesions, respectively, were located in the middle and central third of the lungs. A bronchus sign was present in 166 (77%) of cases. 

Choice of biopsy tools are presented for each of the operators in Figure 1. Operator 1 and 2 primarily used a combination of forceps, cytobrush and bronchial lavage, whereas operator 3 and 4 primarily used only forceps.

Fluoroscopy was used in 197 of the 215 procedures (92%). During these procedures, the lesions were visible in 147 (75%) of cases. rEBUS was used in 206 of 215 procedures (96%). Operators’ reasons for not using rEBUS were termination of the procedure prior to biopsy sampling due to unsuccessful navigation to the lesion (n = 7) or rEBUS not working (n = 2). The lesions were visible on rEBUS in 154 (75%) cases.

The univariate comparisons between lesion and procedure characteristics and diagnostic yield are presented in Table 2. Diagnostic ENB procedures were associated with a minor yet significant difference in tumor size compared to non-diagnostic ENB procedures (28 mm vs. 24 mm, *p* = 0.03). When the lesion was visible on fluoroscopy, the procedure was more likely to be diagnostic when compared with lesions that were not visible *p* = 0.003, and likewise, when the lesion was visible on rEBUS, the procedure was more likely to be diagnostic when compared with lesions that were not visible *p* = 0.001. There were no significant associations between lesion location (peripheral, middle or central) and diagnostic yield (*p* = 0.71) or between presence of a bronchus sign and diagnostic yield (*p* = 0.27).

A cross-classified random effects logistic regression model was used to assess the effects of each operator’s progressive ENB experience (=repetition number) on diagnostic yield *while adjusting* for lesion and procedure characteristics, such as lesion size and location, presence of a bronchus sign and lesion visibility on fluoroscopy and rEBUS (Table 3). In this model, lesion visibility on fluoroscopy, but not repetition number, presence of a bronchus sign or lesion size or location were found to impact diagnostic yield (Table 3).

## 4. Discussion

In this multicenter study of 215 consecutive ENB procedures performed by four operators, the multivariate regression analysis with adjustment for factors with suspected association to diagnostic yield showed that lesion visibility on fluoroscopy, but not visibility on rEBUS, nor lesion size or location or operators’ repetition number, predicted diagnostic yield. Our study is the first to evaluate predicting factors of diagnostic yield in ENB in novice ENB operators including data from operators’ very first ENB procedures.

During the operators´ first year of ENB performance, the repetition number of the procedures was not associated with diagnostic yield. This supports that ENB is a complicated procedure and that competency is not obtained during operators’ initial 50–75 procedures. This underlines the need for a structured approach simulator training to shorten operators’ learning curves and ensure basic competency prior to unsupervised performance. To support this, a randomized trial found that simulation-based training was more efficient than traditional apprenticeship training in the beginning of novice EBUS operators’ learning curves [16]. Furthermore, earlier studies evaluating EBUS and ENB have shown that the learning curves of individual trainees vary considerably [14,17] and international guidelines now recommend the use of structured assessment of performance, instead of relying on an arbitrary number of performed procedures to ensure competency [18]. Furthermore, earlier studies evaluating EBUS and ENB have shown that learning curves of individual trainees vary considerably [14,17] and international guidelines now recommend the use of structured assessment of performance instead of relying on an arbitrary number of performed procedures to ensure competency [18]. It is also important to note that other factors also contribute to the complexity of the ENB procedure, with one major important problem being the challenge of image-to-patient divergence due to (1) the patient usually not being positioned in the exact same position during CT-imaging and on the operation table and (2) the lungs, especially the lower parts, potentially moving during breathing, and thereby, the real-time location of a given target will often not perfectly match the position known from the preoperative CT-scan [2]. 

The observed diagnostic yield in our study was 57% which is low when compared to what has been reported from observational ENB studies during the last decade, where diagnostic yields range from 56–88% [7,19]. This is likely because the majority of studies have exclusively included data from hospitals where operators had extensive experience with ENB, whereas we have focused on operators and hospitals with *no* prior ENB experience.

We found that lesion visibility on fluoroscopy significantly predicted diagnostic yield, which we are the first to have shown. In contrast, lesion location did not impact diagnostic yield in the multivariate analysis, which is in line with what has been reported previously [20,21]. Though lesion visibility on rEBUS was associated with diagnostic yield in our univariate analyses, the association did not remain in the multivariate regression model. Previous studies, including the large-scale NAVIGATE study of diagnostic yield in ENB, also found that use of rEBUS did not increase diagnostic yield [22]. It shall be noted that we did not include registration of details about the view seen on rEBUS, for example, in eccentric versus concentric view. This could be relevant to include in future studies. 

When comparing the overall diagnostic yield across different studies it must be taken into account that the availability to use additional multimodality such as for example ROSE can differ between centers. In the present study, ROSE was not available due to local standards. It can be speculated whether this contributes to the low diagnostic yield. However, we do not believe that this is the case, since a recent meta-analysis reported equal rates of diagnostic yield in ENB studies where ROSE was used, when compared to studies that did not utilize ROSE [23]. 

In our study, all procedures were performed using conscious sedation. It can be speculated whether sedation modality (conscious sedation versus general anesthesia) would have affected our results. We are not aware of results from randomized controlled trials of conscious sedation versus general anesthesia during ENB, however, an RCT of conscious sedation versus general anesthesia during EBUS-guided transbronchial needle aspiration showed that the use of general anesthesia did not result in higher diagnostic yield [24].

All complementary tools, including choice of biopsy tools, were performed at the operator’s discretion. It can be speculated whether a uniform protocol for the use of biopsy tools would have affected the results. As this study was a “real-life setting”, we considered it to be up to each of the operators to evaluate and choose which tool to use, and that this was part of the learning process. It might be questioned why needle aspiration was not used more often by the operators, as it has been demonstrated in a metanalysis that transbronchial needle aspiration has a higher diagnostic yield than transbronchial biopsy [25]. However, it is our belief that operators have—with their choice of biopsy tools—strived to balance a theoretical benefit of transbronchial needle aspiration against the increased risk of complications when using this technique close to pleura. 

In our study, the presence of a bronchus sign did not influence diagnostic yield. Previous studies have shown varying result since some [8] but not all authors [26] found that ENB procedures were more likely to be diagnostic if a bronchus sign was present. A possible explanation may be that the former studies primarily have included peripherally located lesions, whereas most studies, including ours, included central, middle and peripherally located lesions. 

We only observed a few severe complications: pneumothorax requiring chest tube insertion occurred in two cases (0.9%) and bronchopulmonary hemorrhage requiring a balloon catheter occurred during one procedure (0.5%). Previous studies of ENB have reported similar low incidences of adverse events which supports the fact that the procedure is safe even in the hands of novice ENB operators [27]. 

Our study has a number of strengths. First, it is among the first of its kind to evaluate the impact of operators’ progressive case volume on diagnostic yield and to what extent other factors influence outcomes. It is thus of great importance that we were able to include exclusively operators *without* ENB experience and our results thereby reflect real-life situations in hospitals with a recent purchase of ENB equipment. Second, previous studies of diagnostic outcomes of ENB in the hands of inexperienced operators have been single center studies including only one or two operators [28,29,30]. In contrast, the multicenter design of our study strengthens the results and ensures their external validity.

There are limitations to our study which deserves full consideration. First, it can be speculated whether the uniformly lower sampling performance by the novice ENB operators may have blurred a potential significant difference in each predictor (e.g., rEBUS detection, bronchus sign, etc.) if the procedure had been carried out by an experienced operator. Second, due to local and regulatory study deadlines and delays, operators did not end up with an equal number of procedures. Third, there may be factors related to local practice in the three different pulmonary departments that we were unable to control for, for example, the number of assistants in the operating room during the procedures. However, operating rooms and invasive procedures, including endoscopic sampling in public hospitals in Denmark, are in general, uniform and standards are aligned across the country. 

## 5. Conclusions

We found that when ENB is performed by operators with great experience in conventional bronchoscopy and endobronchial ultrasound but no prior experience with ENB, visibility of the lesion on fluoroscopy, but not operators’ progressive ENB experience assessed by repetition number during the first 25–79 procedures, predicts diagnostic yield.

## Figures and Tables

**Figure 1 diagnostics-12-03127-f001:**
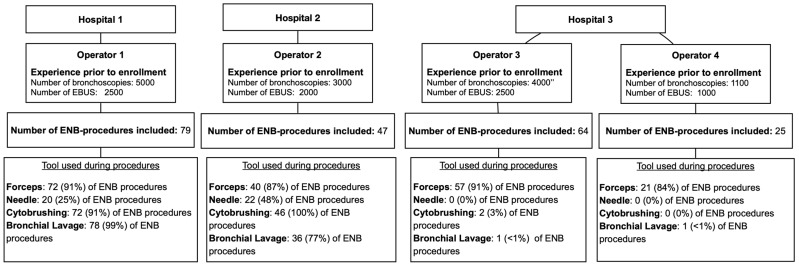
Overview of sites, operators and their choice of biopsy tools. ENB: electromagnetic navigation bronchoscopy.

**Figure 2 diagnostics-12-03127-f002:**
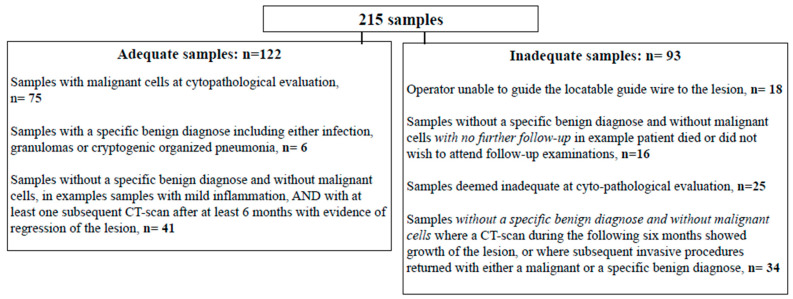
Description of adequate and inadequate procedures.

**Table 1 diagnostics-12-03127-t001:** Procedure and lesion characteristics.

Number of Procedures	215
Lesion size ^§^, mm (mean (SD))	26 (12)
Lesion location (lobe) ^#^	
Right upper lobe	74 (34%)
Right middle lobe	13 (6%)
Right lower lobe	34 (16%)
Left upper lobe	61 (28%)
Left lower lobe	32 (15%)
Lesion location (lung zone)	
Peripheral 1/3 of the lung	126 (59%)
Middle 1/3 of the lung	67 (31%)
Central 1/3 of the lung	22 (10%)
Lesion visible on rEBUS ^§§^	
Lesion visible on rEBUS	154 (75%)
Lesion not visible on rEBUS	52 (25%)
Lesion visibility on fluoroscopy ^##^	
Lesion visible on fluoroscopy	147 (75%)
Lesion not visible on fluoroscopy	50 (25%)
Bronchus sign * present (no. (%))	166 (77%)

SD = standard deviation. rEBUS: radial endobronchial ultrasound; ^#^ n = 214; ^§^ Longest axis; ^§§^ rEBUS was used in 206 of the 215 procedures (96%); ^##^ Fluoroscopy was used in 197 of the procedures (92%); * Bronchus sign: air-filled bronchus in close proximity to the lesion as seen on computed tomography imaging.

**Table 2 diagnostics-12-03127-t002:** Characteristics of diagnostic and non-diagnostic ENB procedures (univariate comparisons).

		Diagnostic Procedure N = 122	Non-Diagnostic Procedure N = 93	Total N = 215	*p* Value
Lesion size, mm (mean (SD))		28 (12)	24 (12)	26 (12)	0.03
Lung zone (no. (% of total))					0.71
	Central (no.)	14 (64%)	8 (36%)	22	
	Middle (no.)	39 (58%)	28 (42%)	67	
	Peripheral (no.)	69 (55%)	57(45%)	126	
Operator (no. (% of total))					0.08
	Operator 1	43 (54%)	36 (46%)	79	
	Operator 2	28 (60%)	19 (40%)	47	
	Operator 3	42 (66%)	22 (34%)	64	
	Operator 4	9 (36%)	16 (64%)	25	
Lesion visible on fluoroscopy ^§^					0.003
	Lesion visible on fluoroscopy	95 (65%)	52 (35%)	147	
	Lesion not visible on fluoroscopy	20 (40%)	30 (60%)	50	
Lesion visible on rEBUS ^$^					0.001
	Lesion visible on rEBUS	98 (64%)	55 (36%)	154	
	Lesion not visible on rEBUS	24 (46%)	28 (54%)	52	
Presence of a bronchus sign *		98 (59%)	68 (41%)	166	0.27

SD = standard deviation; rEBUS: radial endobronchial ultrasound; ^§^ Fluoroscopy was used in 197 of the ENB procedures; ^$^ Radial EBUS was used in 206 of the ENB procedures; ***** Bronchus sign: An air-filled bronchus in close proximity to the lesion as seen on computed tomography imaging.

**Table 3 diagnostics-12-03127-t003:** Predictors associated with diagnostic yield (multivariate comparisons): Cross-classified random effects logistic regression.

	Odds Ratio for a Diagnostic Procedure *	SE	z	*p*	95% CI	
Repetition number ^§^	1.01	0.001	0.64	0.52	0.99	1.02
Lesion size, mm	1.00	0.01	0.28	0.78	0.98	1.03
Lung zone	0.61	0.22	–1.37	0.17	0.31	1.24
Lesion visible on fluoroscopy	3.10	1.33	2.64	0.008	1.34	7.20
Lesion visible on rEBUS	1.9	0.83	1.48	0.14	0.81	4.50
Bronchus sign present ^#^	1.04	0.47	0.10	0.92	0.43	2.54

rEBUS: radial endobronchial ultrasound; SE = Standard error; ***** Cross-classified random effects specified to adjust for Operator and lesion location (lobe); operator random effects SD = 0.36 (SE = 0.32) and lesion location random effect SD = 0.53 (SE = 0.30); ^§^ Repetition number: The number of ENB procedure when counted from each operator’s first ENB procedure (=repetition number 1) and progressively forward. ^#^ Bronchus sign: An air-filled bronchus in close proximity of the lesion as seen on computed tomography imaging.

## Data Availability

The raw/processed data required to reproduce the above findings cannot be shared at this time due to technical/time limitations.

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
