# Peer review of "When Pulmonologists Are Novice to Navigational Bronchoscopy, What Predicts Diagnostic Yield?"

_diagnostics, 2022, doi:10.3390/diagnostics12123127_

Round 1
Reviewer 1 Report
This is a multicenter prospective observational study aimed to clarify predictors for diagnostic ENB procedures by ENB novices. I read this report with great interest, as there had never been a study of this kind before. Points for improvement are described below. I hope this will be helpful.
Major comments
Line 223
Cone-beam CT and augmented fluoroscopy, as a solution for lesions invisible on fluoroscopy, should be discussed, along with a significant clinical problem in ENB: CT-to-Body divergence.
Lines 235-239
The results of the meta-analysis of ENB studies should be emphasized over the old single-center study of conventional fluoroscopic bronchoscopy published 15 years ago.
Lines 240-245
The optimal sedation method should be discussed from multiple perspectives, including not only diagnostic yield but also the use of cone-beam CT.
Lines 266-273
Limitations as well as strengths of this study should be mentioned carefully.
If the diagnostic performance is uniformly low due to sampling errors from being a novice, it will be difficult to find significant differences in each predictor (e.g., rEBUS detection, bronchus sign, etc.).
Minor comments
Methods
Please add details about conscious sedation agents and the type (and manufacturer) of bronchoscope, rEBUS probe, and sampling device.
Was visibility on fluoroscopy confirmed with frontal images (or even oblique images)?
Please provide details.
Line 90
Which EBUS are you referring to, radial or convex probe?
Lines 144-146
Have you confirmed that the continuous variables compared using ANOVA are normal distribution?
Please correct mistypes and spell out abbreviations (e.g., EBUS, ENB, OR, SD, S.E.) in the text, figures, and tables.
In particular, Table 2 is difficult to see due to misalignment.
In Figure 1, there is no need to abbreviate bronchial lavage.
“XX of ENB procedures” need not be repeated.
Reviewer 2 Report
I thank the Authors for their valuable work. I found this paper interesting, exploring a field (i.e., the training of interventional pulmonologists in the diagnostic procedures for peripheral pulmonary lesions) where literature is scarce compared with other procedures like EBUS-TBNA. The analysis of data is complete and the results support the Authors' conclusions. Here I give some suggestions intended to make the manuscript more complete.
Abstract
1. Lines 33-37, the punctuation needs to be revised in order to make the sentences clearer.
2. A brief sentence concluding the abstract is needed.
Introduction
3. Line 46-47, Authors declare that “ENB utilizes an electromagnetic technique”. Please give a better explanation of the technique.
4. It appears useful to add a short paragraph citing some data and works on the training for endoscopic techniques used to diagnose peripheral pulmonary lesions, such as “Chen A, Machuzak M, Edell E, Silvestri GA. Peripheral Bronchoscopy Training Using a Human Cadaveric Model and Simulated Tumor Targets. J Bronchology Interv Pulmonol. 2016 Jan;23(1):83-6” and “Livi V, Barisione E, Zuccatosta L, Romagnoli M, Praticò A, Michieletto L, Mancino L, Corbetta L. Competence in navigation and guided transbronchial biopsy for peripheral pulmonary lesions. Panminerva Med. 2019 Sep;61(3):280-289”.
Methods
5. The Authors stated that the indication for ENB “was based on consensus between at least two doctors”. Were these doctors pulmonologist/pulmonary endoscopists? Please give explanation.
6. The Authors stated that “The inclusion period was from May 1st 2018 to February 1st 2020. If an operator had not performed at least 50 ENB procedures by the end of the inclusion period, we extended the inclusion period with an additional 6 months for the operator in question” but in the result section 2 operators (n. 2 and 4) performed less than 50 procedures. Please give explanation.
7. Line 103, correct “centrale” with central.
8. Line 104, avoid the word “radial” as already present in the abbreviation rEBUS.
9. Paragraph 2.6 needs to be revised: I suggest to merge this paragraph with the previous (2.5 Diagnosis and definition of diagnostic yield) describing the features of diagnostic and non-diagnostic procedures with or without bullet lists (as Authors consider more adequate) within this paragraph and using the same text formatting.
Results
10. Figure 2 is cited without explanation.
11. Table 2 is cited before table 1.
Discussion
12. Line 213, merge citation 11-14.
13. The Authors declare that “we have yet no evidence from randomized trials that general anesthesia gives better results than conscious sedation” in lines 243-245. Please better discuss this important point, maybe citing the existing evidences for other techniques (Casal RF,Lazarus DR, Kuhl K, Nogueras-Gonzalez G, Perusich S, Green LK, et al. Randomized trial of endobronchial ultrasound-guided transbronchial needle aspiration under general anesthesia versus mod- erate sedation. Am J Respir Crit Care Med. 2015;191:796–803) and the possible differences and implication with the technique described in this work.
14. The Authors state that “It might be argued that needle aspiration could have 250 been used more often by the operators and could have increased the diagnostic accuracy”. Please provide reference such as “Mondoni M, Sotgiu G, Bonifazi M, Dore S, Parazzini EM, Carlucci P, Gasparini S, Centanni S. Transbronchial needle aspiration in peripheral pulmonary lesions: a systematic review and meta-analysis. Eur Respir J. 2016 Jul;48(1):196-204”.
15. Please give a brief description of the work’s limitation.
Figures and tables
16. Tables need to be revised, and in particular table 2 needs to be redesigned in order to make it more readable and clearer.
Round 2
